# The Effects of Face Coverings on Perceived Exertion and Attention Allocation during a Stepping Task

**DOI:** 10.3390/ijerph19116892

**Published:** 2022-06-04

**Authors:** Robyn Braun-Trocchio, Jessica Renteria, Elizabeth Warfield, Kaitlyn Harrison, Ashlynn Williams

**Affiliations:** Department of Kinesiology, Texas Christian University, Fort Worth, TX 76129, USA; j.d.renteria@tcu.edu (J.R.); e.warfield@tcu.edu (E.W.); k.p.harrison@tcu.edu (K.H.); a.williams5@tcu.edu (A.W.)

**Keywords:** face masks, coronavirus, COVID-19, RPE, physical activity

## Abstract

The COVID-19 pandemic has impacted the entire world from lockdowns to various recommended restrictions including social distancing and wearing face coverings. In a safe environment, cardiovascular exercise is important for both physical health and mental health. The current study examined the effects of face coverings on rating of perceived exertion and attention allocation during an exertive stepping task. Participants completed a stepping task with a weighted vest at 20% of their bodyweight until volitional fatigue with a face covering (*n* = 23) or without a face covering (*n* = 31). Results revealed a non-significant difference (*p* = 0.25) in the duration of the stepping task (in seconds) between the no face covering (*M* = 455.81, *SD* = 289.77) and face covering (*M* = 547.83, *SD* = 285.93) conditions. Results indicated increases in perceived exertion (*p* < 0.001) and heart rate (*p* < 0.001) as time progressed across the four time points (i.e., 30 s, 1/3 time to exhaustion, 2/3 time to exhaustion, and exhaustion) in both conditions. No significant differences were found between the conditions for RPE (*p* = 0.09) and heart rate (*p* = 0.50). Participants wearing a face covering were more internally focused across the duration of the stepping task (*p* = 0.05). This study has relevance for applied practitioners implementing physical activity interventions that require face coverings.

## 1. Introduction

The COVID-19 disease quickly spread the globe, generating a world-wide crisis leading the World Health Organization [1] to declare it a global pandemic. The pandemic has impacted the entire world from lockdowns to prevention strategies including social distancing, hand hygiene, and the wearing of face coverings. COVID-19 is highly transmissible when one encounters an infected person through small respiratory droplets [2]. In order to minimize transmission, many fitness facilities required that face coverings be worn due to increased respiration volume and greater risk of contagiousness [3].

Exercising in a safe environment is important for both physical and mental health. The World Health Organization [4] recommends 150 min of moderate intensity physical activity (PA) or 75 min of vigorous PA a week to reduce the risk of developing chronic diseases such as cardiovascular disease and type II diabetes. PA also alleviates the negative mental health effects including anxiety, depression, and stress [5]. However, maintaining adequate PA during the pandemic has been especially challenging and caused a decline in PA which may have negatively impacted mental health [6]. According to Karageorghis et al. [7], PA reduced by ≈2000 steps per day from pre-lockdown to during the lockdown. Additionally, a decrease in mental health scores was observed. Werneck et al. [8] found that higher physical inactivity was associated with increased anxiety and depressive symptoms during the COVID-19 pandemic. To better cope with the COVID-19 pandemic, maintaining physical fitness is important for both good physical health and mental health [9].

Specifically, the COVID-19 mitigating strategies have led to a reduction in PA and increased sedentary behavior [6,10]. Due to the universal recommendations for wearing face coverings, researchers began to investigate the physiological impacts during exercise, and some researchers report only minor physiological changes [10,11,12]. More specifically, Epstein et al. [10] reported no significant differences in terms of heart rate (HR), respiration rate, blood pressure, oxygen saturation with no mask, wearing a surgical mask and N95 mask within healthy adult participants. However, some researchers reported a reduction in exercise time, decease in maximal oxygen consumption (VO_2max_) and reduced ventilation [13]. There are also mixed results as to whether wearing a face covering impacts HR while exercising. Some research reported that slightly higher HR [14] compared to other research who reported lower levels [13,15], and some researchers report no changes [10,12]. Decreased HR levels could be influenced by the early termination of exercise, while increased HR may be due to increases in ventilation.

Additionally, scholars have examined how face coverings impact the rating of perceived exertion (RPE) but report inconsistent findings [10,12,13,16]. In one study examining the impact of face masks on RPE and cognitive function, Slimani et al. [16] reported that the face mask condition had significantly higher RPE compared to no face mask following a randomized 15-min warm-up protocol. Moreover, the face mask condition in the research of Slimani et al. [16] experienced significantly higher increases in concentration performance compared to no face mask condition. Although non-significant, the face mask condition had fewer errors performing the concentration task. However, this study did not investigate the impact of RPE throughout the task. Moreover, Driver and colleagues [13] examined the impact of wearing a cloth face mask on exercise performance, physiological measures, and perceptual measures of effort and found that the RPE was significantly higher wearing a mask during the first four stages of a maximal cardiopulmonary exercise tests (CPETs) on a treadmill compared to no face mask. However, there were no significant differences in RPE between the two conditions at the point of exhaustion. Contrary to these findings, Shaw et al. [12] and Epstein [10] found no significant differences in RPE between face mask conditions and no face mask during a cycle ergometry exercise protocol. These previous studies have not examined the impact of attention allocation in conjunction with RPE while wearing a face covering.

Association (e.g., attending to breathing or how their muscles feel) and dissociation (e.g., noticing the scenery or listening to music) are two board attentional strategies for coping with the acute sensations during exercise [17]. Attention is represented along a continuum with association and dissociation on opposite ends. More specifically, by monitoring the psychological cues, associative strategies allow an adjustment of effort while dissociative strategies reduce perceptions of exertion and fatigue by diverting attention away from the physiological cues [18,19,20,21]. These strategies involve thoughts that will affect behavior. Across PA contexts, individuals tend to shift attention as a function of increased physical workload [17]. As the intensity of the physical workload increases, individuals tend to alter their attentional focus from a dissociative style to an associative one.

Shifting attentional focus toward the body (association), such as wearing a face covering, instead of environmental stimuli (dissociation), may increase one’s perceived level of exertion [22]. The current study examined the effects of face coverings on RPE, attention allocation, and HR during an exertive stepping task. We hypothesized that individuals wearing a face covering will have increased associative attention allocation due to the face covering diverting their attention internally, which will cause an increase in RPE compared to those without a face covering. Additionally, based on Epstein et al. [10], we hypothesized that no HR differences would be reported between the two groups since the participants complete the stepping task to exhaustion.

## 2. Materials and Methods

### 2.1. Participants

G*Power 3 [23] was used to conduct an a priori power analysis to determine the number of participants required for the study. Using Cohen’s d = 0.25, α = 0.05, with two groups (e.g., no face covering and face covering) and four measures (i.e., 4-time intervals), the respective sample was set at 24. 

Participants were recruited from a university in the south–central area of the U.S. using email announcements, announcements made in classes, posts to social media, and word of mouth. In order to participate, only healthy individuals who did not have any physical or psychological disabilities that would interfere with the completion of an exertive stepping task or contraindications to exercise were included in the study as assessed by the General Health and Life Type Questionnaire—Shortened Version (GHLQ). A total of 54 participants (females = 38: height 168.83 ± 7.18 cm, weight 63.28 ± 9.38 kg and males = 16: height 180.47 ± 7.39 cm, weight 88.36 ± 17.20 kg) between the ages of 18 and 50 years of age (21.19 ± 5.51) completed the study. Participants were either wearing a face covering (*n* = 23) or not wearing a face covering (*n* = 31). 

### 2.2. Physical Task

A stepping protocol adapted from the Young Men’s Christian Association (YMCA) stepping task was used as the exercise regimen [24]. The participants stepped up and down on to the Rogue Resin Plyo Box exercise stepper (Rogue Fitness, Columbus, OH, USA; height = 30.48 cm) in cadence with the metronome at 96 beats per minute. Each participant completed a stepping task with a load corresponding to 20% of individual body weight until volitional fatigue or the participant no longer maintained the cadence as determined by the researcher. The adapted step task was selected because of its ease of administration and is easily learned. Additionally, this is a novel task to most participants, limiting the amount of biases toward the activity, which can influence an individual’s RPE. If participants deviated from the pace, researchers promptly corrected them.

### 2.3. Instruments

#### 2.3.1. Task-Specific Motivation Scale

This is a four-item scale designed by Hutchinson and Tenenbaum [25]. It is designed to examine task-specific self-efficacy (two items based on Bandura’s [26] self-efficacy measurement guidelines), task-specific perceived ability, and task-specific motivation. Participants rated their task-specific self-efficacy, perceived ability, and motivation on a Likert-type scale ranging from 0 (very low) to 10 (very high).

#### 2.3.2. Commitment Check

This scale measured participants’ commitment level and effort investment in the task. Participants were asked to report their commitment and effort investment on a 10-point Likert-type scale at the end of the task. The three items of the scale include: (a) “How hard did you try while you were completing this task?” (b) “How well do you believe you handled any physical discomfort or pain during the task?” and (c) “How much effort did you invest in the task?” Participants rated each item on a Likert-type scale ranging from 1 (none/not at all) to 10 (very much/very well).

#### 2.3.3. Rating of Perceived Exertion (RPE)

The scale measured perceived exertion during a task [27]. The RPE is a 15-point category-ratio scale ranging from 6 (very, very light (rest)) to 20 (exhausted). The higher the RPE score, the higher the rating of perceived exertion.

#### 2.3.4. Attention

A 10-point scale ranging from 0 (external thoughts, daydreaming, environment, singing songs) to 10 (internal thoughts, how body feels, breathing, muscles) was used to measure attention throughout the task performance [28]. The scale was originally designed to represent the continuum of attention strategies ranging from 0 (pure dissociation) to 10 (pure association).

### 2.4. Procedure

IRB approval was granted. Participants refrained from PA for at least 24 h prior to the testing. Data collected occurred from 9:00 to 15:00. Participants in the face covering condition were instructed to wear their own double ply face covering that was comfortable and would stay over their nose and mouth at all times. Their own face covering was worn to limit the implications of wearing something new and potentially moving during the task. Participants were tested individually in the same private research laboratory to reduce the possible social facilitation effect. Participants in the face covering condition wore their face covering throughout the entire protocol. 

First, participants were given an informed consent form outlining the purpose, format, and tasks of the study. Upon agreement to participate, participants completed a demographic and GHLQ to determine if they are a viable candidate. Participant’s height and weight were measured. Then, a Polar H10 HR monitor (Polar Electro Oy, Kempele, Finland) was attached to the participant, and baseline HR measurements were collected for three minutes. After baseline HR, the RPE and attention scales were explained using a standard script. If the participant was unsure about any of the scales, the researcher offered clarification. 

After the instruments were introduced to the participants, they were familiarized with the stepper and the metronome and provided with the opportunity to practice the task. The participants stepped up and down on to the stepper in cadence with the Steinway and Sons Metronome App from Apple at 96 beats per minute. Familiarization included the participant taking eight unweighted steps paced by a metronome while connected to an HR monitor. The task-specific motivation scale was administered after the participants felt comfortable with the task.

Next, the participant was fitted with the Rogue Fitness MiR weighted vest (Rogue Fitness, Columbus, OH, USA) corresponding to a load of 20% of the participant’s body weight. Prior to the task, HR was collected by the researchers and, the participants verbally stated their RPE and attention to determine baseline measurements. Participants then completed the adapted YMCA stepping task [24] with the weighted vest until volitional fatigue or the participant no longer maintained the cadence. RPE and attention were collected verbally from the participants at 30 s intervals throughout the task. HR was also collected at 30 s intervals by the researchers. Upon completion of the task, participants completed a commitment check and were debriefed.

### 2.5. Data Analysis

The Statistical Package for the Social Sciences (SPSS version 26, SPSS Inc., Chicago, IL, USA) was used to analyze the data. Multivariate Analysis of Variances (MANOVA) was utilized for the motivation and commitment check items. An independent t-test measured task duration. To test the hypotheses, a repeated measure Analysis of Variance (RM ANOVA) was used for HR, RPE and attention allocation through increment of physical effort expenditure. The two conditions (i.e., no face covering condition and face covering condition) were considered the between-subject factors, and time interval (i.e., 30 s, 1/3 time to exhaustion, 2/3 time to exhaustion, and exhaustion) was considered the within-subject repeated factor. When Mauchly’s sphericity reached significance for the main effects (*p* < 0.05), then the Greenhouse–Geisser (GG) epsilon correction coefficient was implemented. Partial eta squared (*η_p_*^2^) was used as a measure of effect size.

## 3. Results

### 3.1. Task Specific Motivations

To test pre-task differences in motivation between the two conditions (i.e., no face covering condition and face covering condition), a MANOVA was conducted. A non-significant condition was revealed Wilk’s λ = 0.95, *F* (4, 49) = 0.67, *p* = 0.61, *η_p_*^2^ = 0.05. Participants were highly motivated (*M* = 8.70, *SD* = 0.18).

### 3.2. Task Duration

To measure the extent to which the face covering affected task adherence, an independent sample t-test was performed. Results revealed a non-significant difference (*p* = 0.25) between the lengths of time in the two conditions. Specifically, the mean time duration (in seconds) for each group was no face covering condition *M* = 455.81, *SD* = 289.77 and face covering condition *M* = 547.83, *SD* = 285.93.

### 3.3. Commitment Check

Task commitment was measured by a MANOVA between the two conditions. A non-significant condition was revealed: Wilk’s λ = 0.91, *F* (3, 50) = 1.57, *p* = 0.21, *η_p_*^2^ = 0.09. In general, participants were committed to the task (*M* = 4.60., *SD* = 0.21)

### 3.4. Heart Rate (HR) Measure

A RM ANOVA with time interval (i.e., 30 s, 1/3 time to exhaustion, 2/3 time to exhaustion, and exhaustion) was considered a within factor and the conditions (i.e., no face covering condition and face covering condition) as the between factor was conducted for HR in beats per minute (BPM) (see Figure 1). A significant main for time interval was revealed, GG_ms_ = 44,684.25, *F* (1.34, 69.42) = 413.09, *p* < 0.001, *η_p_*^2^ = 0.88, indicating that HR increased over time in both conditions. The main effect for the conditions, *F* (1, 52) = 0.46, *p* = 0.50, *η_p_*^2^ = 0.01, and the time by condition interaction, GG_ms_ = 18.72, *F* (1.34, 69.42) = 0.17, *p* = 0.75, *η_p_*^2^ = 0.003, were non-significant.

### 3.5. Ratings of Perceived Exertion (RPE)

A four (time intervals listed above) by two (conditions) RM ANOVA was conducted for RPE. A significant main effect for time was revealed, GG_ms_ = 981.49, *F* (1.85, 96.17) = 298.81, *p* < 0.001, *η_p_*^2^ = 0.85, indicating that as time progressed, participants’ RPE increased the conditions (see Figure 2). No significant effect was found for the conditions, *F* (1, 52) = 3.04, *p* = 0.09, *η_p_*^2^ = 0.06. Lastly, the condition by time interaction was non-significant, GG_ms_ = 2.86, *F* (1.85, 96.17) = 0.86, *p* = 0.42, *η_p_*^2^ = 0.02.

### 3.6. Attention Allocation

The analysis conducted on attention allocation was similar to the one conducted on RPE. A significant main effect of time was revealed, GG_ms_ = 286.25, *F* (1.62, 84.20) = 71.64, *p* < 0.001, *η_p_*^2^ = 0.58, with participants’ attention becoming more internal over time across the conditions. The condition effect was significant, *F* (1, 52) = 4.00, *p* = 0.05, *η_p_*^2^ = 0.07. The participants wearing a face covering were more internally focused across time (see Figure 3). The condition by time interaction was non-significant, GG_ms_ = 0.38, *F* (1.62, 84.20) = 0.09, *p* = 0.87, *η_p_*^2^ = 0.002.

## 4. Discussion

The primary purpose of this study was to examine the effects of face coverings on perceived exertion and attention allocation during an exertive stepping task. Participants in the different conditions did not differ in RPE during the stepping task, which aligns with previous research from Shaw et al. [12] and Epstein et al. [10]. These results are different than some previous research demonstrating that the face masks had significantly higher RPE compared to no face mask following a warm-up protocol [16]. The current study had participants exercise until exhaustion, completing one task collecting RPE throughout, while Slimani and colleagues [16] had participants complete a warm-up comprised of multiple tasks (e.g., light tempo runs, arms circles, high knees jog, back kicking, and stretching exercises) and collected RPE before and after each session. These differences could have impacted the current findings. Furthermore, Driver and colleagues [13] found higher levels of RPE while wearing a cloth face mask in the first four CPET stages. At exhaustion, RPE was similar between the two conditions similar to the current study. 

The current findings are similar to previous literature with work-related light exercise which revealed that physiological strain and thermal discomfort did not significantly differ between prolonged face mask wearing group and the control group [29]. Furthermore, Garra et al. [30] reported no differences between wearing a medical face mask or the combination of a medical mask and N95 mask on physical fatigue in hospital-based participants. In line with previous research, the participants’ RPE increased linearly with time and effort [25,31]. Similar to Driver et al. [13], the exertive task was terminated when RPE had reached a high level in both conditions. Moreover, the current findings found no significant differences in HR between the two conditions, which aligns with some previous literature [10,12]. These results are in contrast to research that reported slightly higher HR [13,14] and to other researchers who reported lower levels [15] while wearing a face covering. 

In addition, the current findings revealed that participants wearing a face covering were more internally focused throughout the task. Previous research found that sensory stimuli (e.g., visual, auditory, and olfactory) have a diverting effect and delay the attention shift from dissociative to association [7,26]. In the current study, however, the stimuli of a face covering produced the opposite effect. Since the face covering is on the participant, this may have influenced the attention allocation to be more internal. Previously, Ritchie et al. [31] found that individuals had more associative attention while wearing a flavored mouthguard during an exertive stepping task. In addition, research has demonstrated that face coverings impact perceived dyspnea or difficulty breathing, claustrophobic, and uncomfortable while exercising, especially at higher intensities [13,29], which could result in the current participants having a more internal attention allocation. Despite having more associative attention allocation wearing a face covering, especially during the first two time points, task duration was not significantly different from not wearing one. Our results support previous research [10,12] in terms of no differences being reported on time to exhaustion between wearing a face covering and a control condition.

### Limitations and Future Research

There are several limitations in the present study that should be considered when interpreting the findings. First, this was a between-subject experiment, meaning the participants only completed one condition. While the individual variability presents concern in any between-subject protocol, to control for variability, two manipulation checks (i.e., task-specific motivation scale and commitment check) were employed. Having knowledge of the purpose of the study could have influenced the results in terms of task duration. Future research should consider utilizing a within-subject design to confirm the current findings. Additionally, participants were able to wear their own face coverings, which assists with the fidelity but could have impacted the results. Standardizing the face covering in the future could assist with this limitation and allow participants to become familiarized with the face covering prior to the exertive task. Furthermore, there was an unequal number of males and females in the sample, which may have impacted the results. Only one type of exertive task was utilized, limiting the generalizability across other exertive tasks. Therefore, other exertive tasks (i.e., running, cycling, and resistance training) should examine the effect of face coverings on attention allocation and perceived effort. Finally, the time of day of exercise, environmental conditions (i.e., air temperature, humidity, etc.), and previous experience to a stepping task or rhythmic activities should be considered in the future. 

## 5. Conclusions

This study indicated that while performing an exertive stepping task, individuals wearing a face covering are more internally focused on their body; however, the face covering did not impact the duration of the stepping task, perceived exertion, or HR. The current findings have relevance for applied practitioners implementing PA interventions that require face coverings. Practitioners can inform clients that face coverings do not impact RPE, HR, or task duration and design exercise programs accordingly. The COVID-19 pandemic has increased the rate of sedentary activity, physical inactivity, and impacted mental well-being. Therefore, it is important to help promote active and healthy lifestyles, while wearing a face covering, more than ever for physical health and mental health. 

## Figures and Tables

**Figure 1 ijerph-19-06892-f001:**
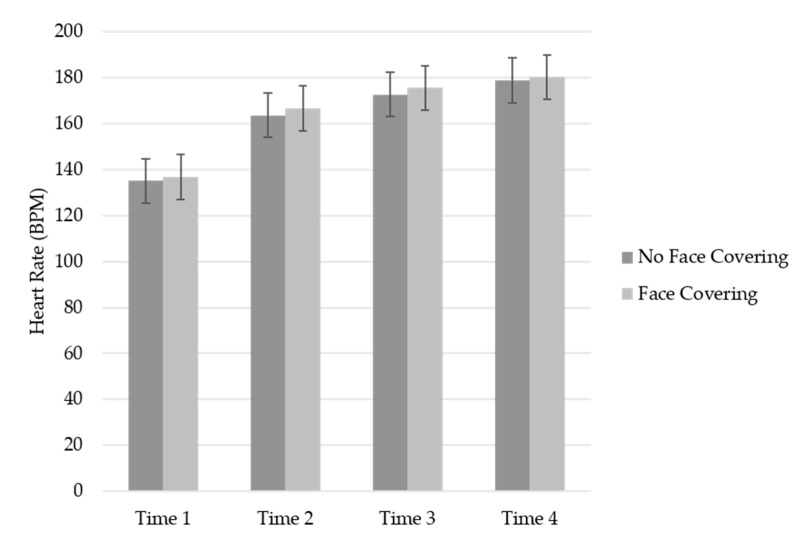
Mean HR across time by condition.

**Figure 2 ijerph-19-06892-f002:**
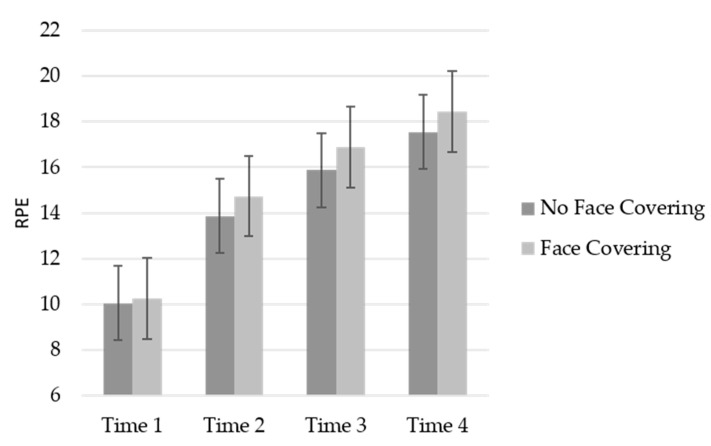
Mean RPE across time by condition.

**Figure 3 ijerph-19-06892-f003:**
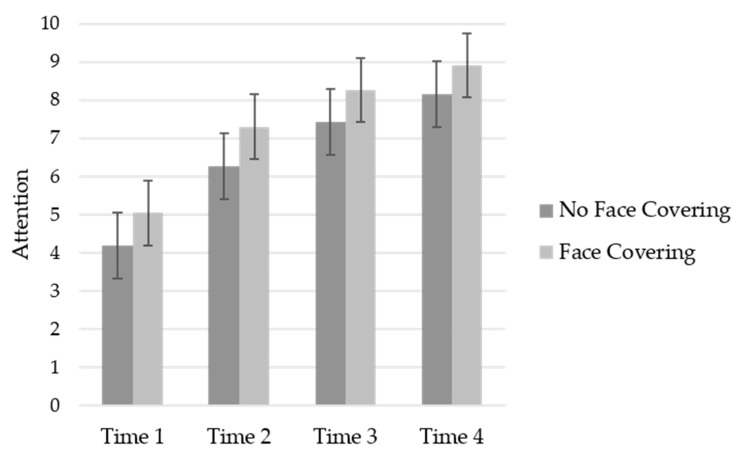
Mean attention across time by condition.

## Data Availability

The data presented in this study are available on request from the corresponding author.

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
