# Peer review of "The Effects of Face Coverings on Perceived Exertion and Attention Allocation during a Stepping Task"

_ijerph, 2022, doi:10.3390/ijerph19116892_

Round 1

Reviewer 1 Report

Thank you for the opportunity to review this paper on the effects of face coverings during exercise. Overall I believe this paper has merit, however there are a few areas I would like to see additional information or clarification. Line 38 indicates the pandemic has caused a decrease in PA and negatively impacted mental health. While I don’t disagree, it is not clear if you are making the claim that the PA decline is what is driving the mental health concerns, or if they are separately impacted by COVID. It would be important to clarify this point. In the methods (lines 98-104) it is not stated whether the gender proportion is roughly equal between the two groups. Given the big gender discrepancy in the total sample, I think this is an important thing to note. Also, is there any literature that discusses differences in perceived exertion or perceived impact of mask wearing between genders? If so, this needs to be stated given your gender breakdown. Also in the methods (line 150), it is noted that height and weight were measured, but no details on how that was done. Those are important parameters to provide the specific details. In the results (line 212-214) it states ‘slightly higher RPE’ in one of the groups, but this is not a statistically significant finding, therefore I am unclear what it is stated in this way. Conversely, duration is also not statistically significant between groups, however the raw scores between the groups show one much higher than the other, yet there was no mention of ‘slightly higher duration’. My concern here is that I am not sure the data are being presented objectively. Further, in the conclusion (lines 281-283), it states ‘face coverings did not impact pe or HR, but nothing is mentioned about duration. It seems the variable of duration is looked at differently by your team, so I think you can address all the variables in a much more objective manner. Finally, lines 285-86 again allude to a relationship between increasing sedentary and decreasing mental health. As noted earlier, this needs to be clarified. 

Finally, throughout the paper there are opportunities to improve the writing, so I suggest that be attended to. For instance, lines 70, 91-92, 164, 201, all have noticeable typos.

Reviewer 2 Report

The Covid-19 pandemic has impacted various recommended restrictions such as face coverings. PA is very important for maintaining of the physical and mental health. It is interesting and important to know how face coverings effect psychological and physiological parameters of exercising persons. Do they allow to receive the positive influence on health after the PA.

The manuscript by Robyn Braun-Trocchio, et al reveals some psychological and physiological differences under face covering and non-face covering conditions during PA. It is written rather clearly, however this some inaccuracies. The research itself is quite short and limited in my opinion. However, due to the novelty of problem it may be published as preliminary findings.

Major comments:

  1. First of all, I would recommend to specify the title due to the limitations such as only one type of physical task applied. The current title suits more for a comprehensive research.
  2. The abstract should be rewritten. It says that results indicated increases in RPE (p < .001) and heart rate (p < .001) along four time points which contradicts to Results and Conclusions sections.
  3. This is not clear how the RPE and attention were indicated during the task (at 4 time points). Was it a subjective rating (by the participants or researchers) or not? How was the measures implemented? Please, provide a sufficient description.
  4. In my opinion it is important to take into account lifestyle of the participants. Are they athletes or may be some of them practiced Irish dance in the past. It could influence the results. Also, the influence of biorhythms is very important. What is the time of the experiment (morning, evening)? Also, external conditions such as air temperature, humidity and illumination could influence the results. Were they the same for all the participants? In my opinion the research is insufficient at the present. It should at least include several samples with the similar conditions to draw conclusions.
  1. The difference in motivation between two conditions is not shown, however it is said that it was determined. Did you find any difference?

Minor comments:

  1. Line 9: Please, specify what kind of exercising do you mean.
  2. Lines 14-15: Length of time of what?
  3. Lines 16 and 17: Did you find increase for face covering or non face covering conditions? It is not clear.
  4. Line 30: The statement that COVID-19 penetrates the lungs during PA here is rather controversial. Do direct penetration of COVID-19 to lungs correlates with more severe symptoms or a higher probability of infections? Please, provide an appropriate reference. What is the sense of the phrase?
  5. What is VO2max, YMCA? Please, provide the meanings in the text when mentioned first.
  6. Line 50: Did you mean decrease?
  7. The hypothesis that no HR differences would be reported between groups is not quite clear. Please, explain the background of your hypothesis.
  8. Please, provide the company name and manufacturer country in the brackets for all the used equipment.
  9. Some increase in time duration for the participants with face covering is unexpected. May be they tried more due to the bold pre-formalization with the purpose of the study?
  10. Line 209: Please, specify that you consider four time intervals listed above.
  11. Line 265: Did you mean face covering?
  12. I suggest to indicate the Limitation paragraph separately.

Reviewer 3 Report

Comments to manuscript ID ijerph-1713598 entitled: The Effects of Face Coverings on Perceived Exertion and Attention Allocation. This manuscript analyzed the influence of wearing face mask in the exercise development. Grammar mistakes must be correct in the manuscript.

0.- Abstract

- Line 9: Please revise and correct grammar mistakes

- Authors must erase acronyms in abstract as RPE

- Are keywords MeSH terms?

1.- Introduction

-Lines 52-52: Please add reference

-Lines 57-58: Please add reference

2.- Material and methods

- Line 144: All the participants wore the same kind of mask? Could the results be different?

- Did authors used any statistical software analysis.

- Can authors add inclusion and exclusion criteria in the research?

3.- Results

Are clearly exposed

4.- Discussion

- Lines 232 and 243: Please correct grammar mistakes

- Line 268: Can authors explain better what means “between-subject protocol”?

5.- Conclusion

OK

Round 2

Reviewer 1 Report

I have now reviewed the revised manuscript and author letter, and do feel the changes that were made are sufficient for publication.

Reviewer 3 Report

Greetings authors for the changes performed in the manuscript. 

Only minor change in line 313: correct the grammar mistake COIVD-19.

Congratulations for the manuscript

This manuscript is a resubmission of an earlier submission. The following is a list of the peer review reports and author responses from that submission.